# Stability of Flavonoid, Carotenoid, Soluble Sugar and Vitamin C in ‘Cara Cara’ Juice during Storage

**DOI:** 10.3390/foods8090417

**Published:** 2019-09-16

**Authors:** Qi Lu, Lu Li, Shujin Xue, De Yang, Shaohua Wang

**Affiliations:** 1Institute of Agro-Products Processing and Nuclear Agricultural Technology, Hubei Academy of Agricultural Sciences, Wuhan 430064, China; luqihzau@126.com (Q.L.); Lulilu2662@163.com (L.L.); xsj2000@163.com (S.X.); yde0537@163.com (D.Y.); 2College of Food Science and Technology, Huazhong Agricultural University, Wuhan 430070, China

**Keywords:** ‘Cara Cara’ juice, storage, hydrophilic and lipophilic antioxidant, carotenoid, flavonoid, degradation

## Abstract

In view of understanding the stability of sterilized ‘Cara Cara’ juice during storage, the changes of specific quality parameters (flavonoid, carotenoid, vitamin C, soluble sugar and antioxidant activities) of ‘Cara Cara’ juice were systematically investigated over the course of 16 weeks in storage at 4, 20, 30 and 40 °C. Total flavonoid and carotenoid indexes showed slight degradation at each temperature, while vitamin C and soluble sugar degraded intensively, especially at 40 °C storage with a great amount of HMF (5-hydroxymethylfurfural) accumulated. There were 29 carotenoids detected during storage, including carotenes and carotenoid esters. Carotenes were kept stable, while the degradations of carotenoid esters were fitted by biexponential function. Carotenoid ester group 2 contained epoxy structures that quickly decreased in the first four weeks at all storage temperatures, while the ester group 1 (belonged to β-cryptoxanthin ester) was degraded gradually. The 13- or 15-cis-lycopene, isomerized from all-(trans)-lycopene, increased with storage time at each temperature. Total flavonoid and carotenoid indexes in stored ‘Cara Cara’ juice were positively correlated with hydrophilic and lipophilic antioxidant abilities.

## 1. Introduction

Citrus juice possesses an attractive natural color, with a sweet and sour taste, making it popular with food consumers around the world [1]. Intake of citrus juice is confirmed to be effective for prevention of human chronic-degenerative diseases [2,3], and micronutrients of carotenoids, flavonoids and ascorbic acid are responsible for the physiological function of citrus juice [4,5]. Orange cv. ‘Cara Cara’, a bud mutation of navel orange (*Citrus. sinensis* L. Osbeck) originating in Venezuela in the 1980s, displays an attractive bright red color due to the accumulation of lycopene [6], and it has been widely planted in China [7]. Changes of food sensorial and nutritional quality during storage limits the date of food consumption. ‘Cara Cara’ juice products have not been commercially available in China, and the nutritional changes of ‘Cara Cara’ juice during storage have not been investigated.

Citrus juice products are usually exposed to various temperatures in the food supply chain. It is necessary to investigate the changes of carotenoid, flavonoid and ascorbic acid during storage at different temperatures, since these components play an important role in the healthy function of citrus juice. Rapisarda reported that the flavanone in sweet orange juice decreased about 50% after storage at 4 °C for 20 days [8], while Klimczak found that the flavanone in commercial pure orange juice was rather stable during storage with only minor changes observed [9]. Apigenin-6,8-di-*C*-glucoside, narirutin-4’-*O*-glucoside, narirutin, hesperidin and didymi were confirmed as typical flavonoids in ‘Cara Cara’ juice [10]. There is little information on the influence of storage temperature and duration on flavonoid content in ‘Cara Cara’ juice.

Due to the complex composition of carotenoids in oranges, the saponification procedure has typically been applied to simplify the analysis, by transferring esterified carotenoids into free cartoenoids [11,12]. The stability of carotenoids can be influenced by temperature, time, and the availability of light and oxygen [13]. Previous studies have mainly focused on the degradation of free carotenoids during storage [12,14], while the change of carotenoid esters in citrus juice during storage has not been reported. There have been 19 carotenoid esters inferred in ‘Cara Cara’ fruit, with the 9-cis-violaxanthin ester confirmed as the dominant component [7], and esterified β-cryptoxanthin considered to be the most stable ester during thermal treatment [15]. The change of free and esterified cartoenoids in ‘Cara Cara’ juice needs to be further explored.

Non-enzymatic browning, frequently observed in citrus juice, plays an important role in the color, flavor and nutritional quality of the stored citrus juice [1]. Non-enzymatic browning is also strongly connected to the degradation of ascorbic acid and sugar, with HMF (5-hydroxymethylfurfural) detected as the indicator [1]. A previous report confirmed that vitamin C in citrus juice was affected by the storage temperature and duration [9]. However, changes of flavonoid, carotenoids, vitamin C and sugar in ‘Cara Cara’ juice during storage at different temperatures have been not investigated, especially in terms of carotenoid esters. 

We, therefore, carried out a pilot study to investigate the changes of carotenoid, flavonoid and vitamin C in ‘Cara Cara’ juice during 16 weeks of storage at different temperatures. In addition, the lipophilic and hydrophilic antioxidant abilities of stored ‘Cara Cara’ juice were analyzed. 

## 2. Materials and Methods

### 2.1. Sample Preparation

‘Cara Cara’ juice (15.3 °Brix, pH 3.70, titratable acidity 0.93%) was obtained from commercial matured fruit of Orange cv. Cara Cara, harvested from the Fujian province of China in December 2017. The fresh ‘Cara Cara’ fruit (50 kg) was immediately peeled and squeezed with a fruit extruder. Half of the crude juice was stored at −80 °C and half was directly subjected to a rapid thermal sterilization. Specifically, ‘Cara Cara’ juice was boiled in a stainless-steel container with an electronic thermometer (F1, invisible, Guangdong, China) monitoring the internal temperature of the ‘Cara Cara’ juice (98 °C, 16 s). The sterilized juice was immediately filled into glass bottles (50 mL). These bottles and their caps were disinfected before use. After cooling, the juice was stored at 4, 20, 30 and 40 °C for 16 weeks without lights. At each sampling time, three bottles of juice were taken and frozen at −80 °C until use.

### 2.2. Chemicals and Reagents

The standards of narirutin, hesperidin, didymin, lycopene, β-carotene, fructose, glucose and sucrose were acquired from Yuanye Bio-Technology Co., Ltd (Shanghai, China). Phytoene and violaxanthin were purchased from CaroteNature (Lupsingen, Switzerland) and Sigma (St. Louis, MO, USA), respectively. ABTS^+^ (2,2’-azino-bis(3-ethylbenzothiazoline-6-sulfonic acid)) and 2,2-diphenyl-1-picrylhydrazyl (DPPH) were purchased from Yuanye Bio-Technology Co., Ltd (Shanghai, China). High performance liquid chromatography purity solvents, including methyl tert-butyl ether (MTBE), methanol and hexane were obtained from Thermo Fisher Scientific (Leicestershire, UK). Other analytical grade chemicals, such as ethanol, hexane and sodium hydroxide were bought from Sinopharm Chemical Reagent Co., Ltd (Shanghai, China).

### 2.3. Extraction of Carotenoid from ‘Cara Cara’ Juice

The extraction of carotenoids from ‘Cara Cara’ juice was performed according to our previous study [15]. Briefly, ‘Cara Cara’ juice (10 mL) was homogenized with 10 mL ethanol/hexane (4:3, *v*/*v*, 0.1% butylated hydroxytoluene) by stirring at 700 rpm for 0.5 h. The mixture was subsequently centrifuged (19,360× *g*, 4 min) to obtain the liquid phase. After extraction of the residue twice, all the liquid phases were combined, and then washed by separatory funnel to collect the non-polar supernatants. The obtained supernatant was evaporated to dryness, then re-dissolved by methyl tert-butyl ether and filtered (0.22 μm polytetrafluoroethylene filter) for the analysis of carotenoids.

### 2.4. Extraction of Flavonoid from ‘Cara Cara’ Juice

Flavonoids in ‘Cara Cara’ juice were extracted based on our previous study, with minor modification [15]. Briefly, ‘Cara Cara’ juice (1 mL) was homogenized with the extract solvent (85% aqueous ethanol containing 0.1% HCl, 4 mL) by an ultrasonic cleaner (KQ-500E, Kun Shan Ultrasound Instrument Co., Jiangsu, China) at 40 kHz for 30 min. The mixture was centrifuged (9680× *g* for 5 min) and filtered through 0.22 μm PTFE filter for further analysis.

### 2.5. Antioxidant Assays

Based on a previous study [16], the antioxidant abilities of hydrophilic and lipophilic extracts in ‘Cara Cara’ juice were measured by DPPH and ABTS^+^ assays. The DPPH activity was evaluated as previously described [10], and the final results were expressed as μmol ascorbic acid equivalent (AAE) per milliliter ‘Cara Cara’ juice (μmol AAE/mL, *y* = 0.0012*x* + 0.0438, *R*^2^ = 0.9902). The ABTS^+^ assay was conducted according to the existing protocol [17], while the final results were expressed as μmol Trolox equivalent (TE) per milliliter ‘Cara Cara’ juice (μmol TE/mL, *y* = 0.3025*x* + 0.076, *R*^2^ = 0.9807). The hydrophilic and lipophilic capacities of stored ‘Cara Cara’ juice were calculated every four weeks (0, 4, 8, 12, 16 weeks).

### 2.6. Analysis of Carotenoids and Flavonoids in ‘Cara Cara’ Juice

The analysis of carotenoids was performed on HPLC (2695 system, Waters Corp., Milford, MA, USA) using a C_30_ reversed phase column (250 × 4.6 mm, 5 μm; YMC, Inc. Wilmington, NC, USA) and flavonoids were separated on the Waters Acquity UPLC system (Waters Corp., Milford, MA, USA) with a BEH C_18_ column (100 mm × 2.1 mm, 1.7 μm). Their chromatographic separation, identification and quantification procedures were conducted based on our previous study [15]. The content of flavonoids in ‘Cara Cara’ juice during storage were analyzed every four weeks (0, 4, 8, 12, 16 weeks), and the sampling times for carotenoids quantification in stored ‘Cara Cara’ juice were set at 0, 2, 4, 6, 8, 12 and 16 weeks.

### 2.7. Ascorbic Acid Measurement 

The ascorbic acid was determined by the titration method, using 2,6-dichlorophenolindophenol dye [18].

### 2.8. Sugar Measurement

Modified from a previous study [19], the soluble sugar in ‘Cara Cara’ juice was determined by HPLC (2695 system, Waters Corp., Milford, MA, USA) with 2414 refractive index detector (Waters, Milford, MA, USA), and an inertsil NH_2_ column (250 × 4.6 mm, 5 μm; Dikma Technologies Inc., Beijing, China) was used for sugar separation. Zinic acetate solution (21.9%, 0.1 mL) and potassium ferrocyanide solution (10.6%, 0.1 mL) were added in ‘Cara Cara’ juice (1 mL) to precipitated proteins. Afterwards, distilled water was added to 2 mL, and centrifuged to obtain the supernatant for HPLC analysis. Mobile phase was acetonitrile/water (75/25) and the HPLC operating conditions were set as: injection volume 20 μL; column temperature 40 °C; detector temperature 40 °C; flow rate 1 mL/min. The contents of sugar in ‘Cara Cara’ juice during storage were detected every two weeks (0, 2, 4, 6, 8, 10, 12, 14, 16 weeks).

### 2.9. Statistical Analysis

All the experiments were conducted in triplicate, and the data were presented as mean ± standard deviation of triplicate independent experiments. One-way analysis of variance (ANOVA) was applied to compare the means, and the differences between the means were analyzed by Duncan’s multiple range tests at a significance level of 0.05. Correlation analysis of the matrix was analyzed by Pearson correlation coefficient (*t*-test). All statistical analyses were processed by IBM SPSS Statistics version 20.0. Carotenoid compounds were quantified in ‘Cara Cara’ juice during the 16 weeks of storage at different temperatures. The data were arranged to have carotenoid components at different temperatures as objects (rows) and storage weeks as variables (columns) and processed by principal component analysis (XLSTAT 2016, Addinsoft, New York, NY, USA). The results were presented with graphs plotting the projections of the units onto the components, and the loadings of the variables. Correlation between variables was evaluated by Pearson’s correlation coefficient [20].

## 3. Results and Discussion

### 3.1. Changes of Flavonoids in ’Cara Cara’ Juice

Based on our previous study [10,15], flavonoids were confirmed to be the dominant phenolic compounds in ‘Cara Cara’ juice, including apigenin-6,8-di-*C*-glucoside, narirutin-4’-*O*-glucoside, narirutin, hesperidin and didymin. The content changes of individual flavonoids are shown in Table 1. Compared with a previous study [21], the contents of narirutin and hesperidin reported in this study were relatively higher. Each individual flavonoid was not significantly changed during storage at 4 °C. Didymin and narirutin were stable with no significant decrease observed at all storage temperatures, while other flavonoids (apigenin-6,8-di-*C*-glucoside, narirutin-4’-*O*-glucoside, hesperidin) were significantly degraded during storage at 20, 30 and 40 °C. The degradation of flavonoids in fruit juice was probably associated with the peroxidase activity, which might not be completely inactive by sterilization [22]. 

HMF eluted with flavonoids on UPLC was only detected at 40 °C storage, with a slow accumulation in the first 12 weeks (*y* = 0.006*x* + 0.012, *R*^2^ = 0.778) and a rapid increase was found in the last four weeks (*y* = 0.071*x* + 0.833, *R*^2^ = 0.871), with the final content reaching 285.74 μg/mL. HMF is generated from the decomposition of vitamin C or sugar degradation, and it is typically used to evaluate the deterioration severity of juice [23,24]. 

### 3.2. The Changes of Carotenoids in ‘Cara Cara’ Juice

#### 3.2.1. Carotenoid Composition

Citrus was reported as a natural carotenoids source [25]. A total of 29 peaks were detected by HPLC-DAD (Figure 1), and they were identified according to our previous studies [7,15]. Peaks 1 to 27 existed in the sterilized ‘Cara Cara’ juice before storage, and they were inferred as mutatoxanthin, zeaxanthin, β-cryptoxanthin, luteoxanthin-C14:0, 9-cis-violaxthin-C18:1, ζ-carotene, violaxthin-C16:0, luteoxanthin-C16:0, β-carotene, unknown ester, 9-cis-antheraxanthin-C16:0, 9-cis-violaxthin-C12:0-C14:0, β-cryptoxanthin-C12:0, β-cryptoxanthin-C16:1, 9-cis-violaxthin-C14:0-C14:0, β-cryptoxanthin-C14:0, 13- or 15-cis-β-cryptoxanthin-C18:1, a mixture of 9-cis-violaxthin-C14:0-C16:0 and 9-cis-violaxthin-C16:0-C18:1, 13- or 15-cis-lycopene, β-cryptoxanthin-C16:0, antheraxanthin-C14:0-C16:0, 9-cis-antheraxanthin-C16:0-C16:0, lycopene, phytoene, cis-phytofluene 1, cis-phytofluene 2 and cis-phytofluene 3, respectively. Peaks 28 and 29 were the newly formed compounds, in trace amounts, during storage at 40 °C for 8 weeks, and they were identified as 13- or 15-cis-β-cryptoxanthin and cis-phytofluene 4, based on their mass spectrometry, elution order and ultraviolet–visible spectra [15]. 

It was confirmed that β-cryptoxanthin esters were more stable than their corresponding free forms [26], whereas epoxy-carotenoid esters were liable to degrade in orange juice, since 5,6-epoxy xanthophylls could be triggered into their 5,8-epoxy counterparts with a trace amount of acid [25,26]. Changes of carotenoids in ‘Cara Cara’ juice could be observed by comparing the HPLC chromatograms before and after the storage. Several epoxy-carotenoid esters (peaks 4, 11, 12, 15, 18, 21 and 22; luteoxanthin-C14:0, 9-cis-antheraxanthin-C16:0, 9-cis-violaxthin-C12:0-C14:0, 9-cis-violaxthin-C14:0-C14:0, a mixture of 9-cis-violaxthin-C14:0-C16:0 and 9-cis-violaxthin-C16:0-C18:1, antheraxanthin-C14:0-C16:0and 9-cis-antheraxanthin-C16:0-C16:0) disappeared completely during the storage. The absence of epoxy-carotenoid esters in ‘Cara Cara’ juice was correlated with its long storage period [25]. Carotenoid esters in ‘Cara Cara’ juice were classified in different groups to simplify their quantification. Briefly, β-cryptoxanthin esters including peaks 13, 14, 16, 17 and 20 were categorized as ester group 1; epoxy carotenoid esters including peaks 4, 5, 7, 8, 11, 12, 15, 18, 21 and 22 were sorted as ester group 2; Peak 10 disappeared after saponification procedure, but its structure could not be inferred by MS fragments and UV-Vis spectra. Therefore, peak 10 was defined as unknown ester and was classified as ester group 3. The content of carotenoids in ‘Cara Cara’ juice during storage are presented in the Appendix A.

#### 3.2.2. Carotenoid Degradation

The total carotenoid index in the sterilized ‘Cara Cara’ juice before storage was 309.06 ± 11.28 μg/mL, and the dominate compound was phytoene (69.78%), followed by total phytofluene (19.98%), carotenoid esters (4.15%), β-carotene (3.39%), lycopene (1.69%), and others (1.01%). The total carotenoids showed a declining trend at all storage temperatures, but their degradations did not reach a significant level. Carotenes of phytoene, β-carotene and lycopene were kept stable during storage, while all-trans-phytofluenes and cis-phytofluenes were irregularly fluctuated. Matrix protection might have been responsible for the stability of the carotenes in ‘Cara Cara’ juice [15,27,28]. Carotenoid ester, especially ester group 2, decreased dramatically during storage, and this might be related to their unstable xanthophyl structure which could be easily isomerized and degraded. The content changes of ester group 1 and ester group 2 in ‘Cara Cara’ juice are presented in Figure 2. Their degradation was fitted by biexponential function (Equation (1)) and the detailed kinetic parameters are presented in the Appendix A.
(1)$$y_{t}=y_{\infty }+A_{1}\exp\left(-\alpha t\right)+A_{2}\exp\left(-\beta t\right)$$

y_t_, is the carotenoid concentration at real time; y_∞,_ is the theoretical concentration of carotenoid at infinite time. A_1_ and A_2_ represent the pre-exponential factors; α and β, are the observed rate constants for fast and slow degradation. Biexponential degradation of carotenoid indicated both irreversible (degraded into volatiles or epoxides) and reversible (isomerization) degradation were involved, and this degradation form was also observed during thermal treatment of carotenoid juices [15,29]. Ester group 2 was quickly decreased in the first four weeks at all storage temperatures (Figure 2), while ester group 1 was degraded gradually. Therefore, the storage time of ‘Cara Cara’ juice could be estimated by combining the degrading rates of ester group 1 and ester group 2 at each temperature, and this issue will be explored in our future research.

#### 3.2.3. PCA Analysis

PCA was investigated to understand the segregation and correlation among carotenoid compounds in ‘Cara Cara’ juice at all storage temperatures. According to the PCA results, three principal components were obtained to account for the total variance. PC1 and PC2 accounted for 69.02% and 15.77% of the total variance, respectively. Carotenoid compounds, decreased with the storage time (Appendix A), were sorted in the same group and they were strongly and positively correlated with PC1 (Figure 3A). Other compounds including 13- or 15-cis-lycopenes (4, 20, 30, 40 ℃), cis-phytofluenes (4, 20, 30, 40 °C), ester groups 3 (4, 20, 30, 40 °C) and ζ-carotenes (30, 40 °C) were classified into the other two groups. The 13- or 15-cis-lycopene, derived from all-(trans)-lycopene by isomerization, was increased with storage time at each temperature, and the contents of 13- or 15-cis-lycopenes (4, 20, 30, 40 °C) were strongly and negatively correlated with PC1. The contents of cis-phytofluene, ester group 3 and ζ-carotene were irregularly changed in ‘Cara Cara’ juice during the overall storage period, and their contents at 4, 20, 30, 40 °C were sorted into different groups. Wibowo et al. proved that ζ-carotene increased during juice storage at different temperatures, while Cortés presented the opposite view [12,30]. In this study, the increase of ζ-carotene was just observed at 40 °C. 

As presented in the PCA score plot (Figure 3B), the carotenoid profiles of ‘Cara Cara’ juice stored at different times were clearly divided into four groups. The sterilized juice at 0 week was grouped in the lower right quadrant, showing a positive correlation with PC1 and a negative correlation with PC2. Similarly, the other three groups (2 and 4 weeks, 6 and 8 weeks, 12 and 16 weeks) were distributed in the different quadrants, indicating that they have different correlations with PCs. Storage time of each group (2 and 4 weeks, 6 and 8 weeks, 12 and 16 weeks) had a similar impact on the change of carotenoid.

### 3.3. The Changes of Soluble Sugars in ‘Cara Cara’ Juice

Changes of soluble sugar are shown in Table 2. The total soluble sugars were gradually decreased with the improved temperature and prolonged storage. The total soluble sugars were decreased by 1.39%, 1.63%, 9.33% and 31.68% respectively when the juice was stored at 4, 20, 30 and 40 °C. It was reported that fructose, glucose and sucrose were greatly degraded in grapefruit juice during its storage at 37 °C for 16 weeks [19], and the loss of soluble sugars might be related to browning reactions. In the present study, sucrose in ‘Cara Cara’ juice was minorly hydrolyzed at 4, 20 and 30 °C, but it was completely degraded at 40 °C, with fructose and glucose being increased. It was shown that the hydrolysis of sucrose was followed by pseudo first-order reaction, and the hydrolysis progress was correlated with acid concentration and storage time [31]. The increased content of glucose and fructose in ‘Cara Cara’ juice stored at 40 °C was not stoichiometrically in line with the hydrolyzed sucrose (Table 2), indicating that hydrolyzate might be partly engaged in Maillard reactions [19].

### 3.4. The Changes of Vitamin C in ‘Cara Cara’ Juice

The percent retention of vitamin C in ‘Cara Cara’ juice during storage is shown in Figure 4, and vitamin C retention decreased with prolonged storage and increased temperature. No significant loss of vitamin C was detected in the ‘Cara Cara’ juice stored at 4 °C for 16 weeks. The concentration of vitamin C decreased by 23.93%, 35% and 69.58% respectively when ‘Cara Cara’ juice was stored at 20, 30 and 40 °C for 16 weeks. The degradation of vitamin C in citrus juice has been widely studied, and the degradation mode has been fitted to the first-order reaction [32,33]. A similar result was found in our study. The presence of vitamin C in citrus juice could protect carotenoids from oxidation, and a lower loss of carotenoid was confirmed in vitamin C fortified juice [34]. 

### 3.5. The Changes of Antioxidants in ‘Cara Cara’ Juice

The antioxidants in ‘Cara Cara’ juice were evaluated in hydrophilic and lipophilic fractions, which represented the antioxidant ability of flavonoids and carotenoids, respectively. It was reported that lipophilic fractions usually displayed much lower antioxidant ability than the hydrophilic fraction in common fruits and vegetables [16,35]. A similar phenomenon was also found in our present study, which might be attributed to the higher content of total flavonoid index than that of the total carotenoid index (Table 1 and Appendix A). The changes in both hydrophilic and lipophilic antioxidant abilities in ‘Cara Cara’ juice during the 16 weeks of storage at different temperatures are shown in Table 3.

The ABTS^+^ and DPPH values for both hydrophilic and lipophilic antioxidants were decreased during storage at different temperatures. After storage for 16 weeks, ABTS^+^ assay values for hydrophilic antioxidants decreased by 13.70%, 14.64%, 13.86% and 7.94% respectively, under 4, 20, 30, 40 °C, while DPPH assay values for hydrophilic antioxidants decreased by 9.34%, 23.36%, 27.10% and 24.76% at corresponding storage temperatures. The ABTS^+^ and DPPH values for lipophilic antioxidants were relatively stable, and their significant decrease was only observed at the end of storage at 40 °C.

A correlation between flavonoid compositions and antioxidant activity in hydrophilic extracts from ‘Cara Cara’ juice was explored. A positive correlation was found between the total flavonoid index and hydrophilic antioxidant ability (Table 4), in accordance with a previous study [36]. Each individual flavonoid was significantly and positively correlated with tested antioxidant activities at *p* < 0.01 level (Table 4). Therefore, flavonoid compounds in ‘Cara Cara’ juice were the important contributions to the hydrophilic antioxidant ability. Correlations between carotenoid compositions and lipophilic antioxidant activities of ‘Cara Cara’ juice are presented in Table 5. Consisting of zeaxanthin, β-cryptoxanthin, lycopene, phytoene, phytofluene, ester group 1, ester group 2 and the total carotenoid index were significantly and positively correlated with the tested bioactivities at *p* < 0.01. β-carotene correlated with ABTS^+^ capacity and DPPH scavenging at *p* < 0.01 level and *p* < 0.05 level, respectively. ζ-carotene, 13- or 15-cis-lycopene, cis- phytofluene and ester group 3 were negatively correlated with the tested bioactivities. The result suggested that zeaxanthin, β-cryptoxanthin, lycopene, phytoene, phytofluene, ester group 1 and ester group 2 were contributors to the lipophilic antioxidant ability of ‘Cara Cara’ juice. 

## 4. Conclusions

The micronutrients in ‘Cara Cara’ juice were investigated during storage at 4, 20, 30 and 40 °C for a period of 16 weeks. Total flavonoid and carotenoid indexes showed slight degradation at each temperature, while vitamin C and soluble sugar degraded intensively, especially at 40 °C storage. Although the total carotenoids were stable at each storage temperature, most carotenoid esters were significantly degraded and fitted by biexponential function. Specifically, the ester group 2 with epoxy structures quickly decreased in the first four weeks at all storage temperatures, while the ester group 1 (belonged to β-cryptoxanthin ester) degraded gradually. The combined degrading rates of the two type of esters might be further applied to estimate the storage time of ‘Cara Cara’ juice. Total flavonoid and carotenoid indexes in stored ‘Cara Cara’ juice were positively correlated with hydrophilic and lipophilic antioxidant abilities. This study provided information on changes of flavonoid, carotenoids, vitamin C and sugar in ‘Cara Cara’ juice during storage at moderate and elevated temperatures, which might be useful for the quality prediction of ‘Cara Cara’ juice during storage.

## Figures and Tables

**Figure 1 foods-08-00417-f001:**
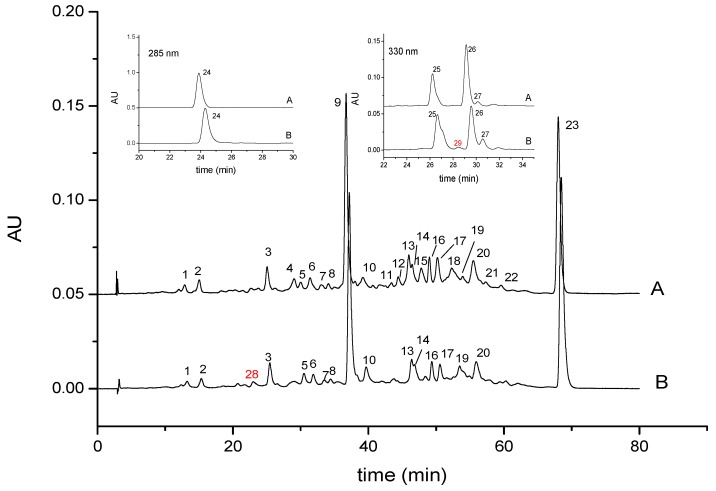
HPLC chromatograms (450 nm) of carotenoids detected in ‘Cara Cara’ juice before (**A**) and after 16 weeks of storage at 40 °C (**B**); Peaks 24–29 displayed no significant absorbance at 450 nm, while distinct absorption was detected at 285 nm (peak 24) and 350 nm (peak 25–29). AU represents the absorbance unit of carotenoid.

**Figure 2 foods-08-00417-f002:**
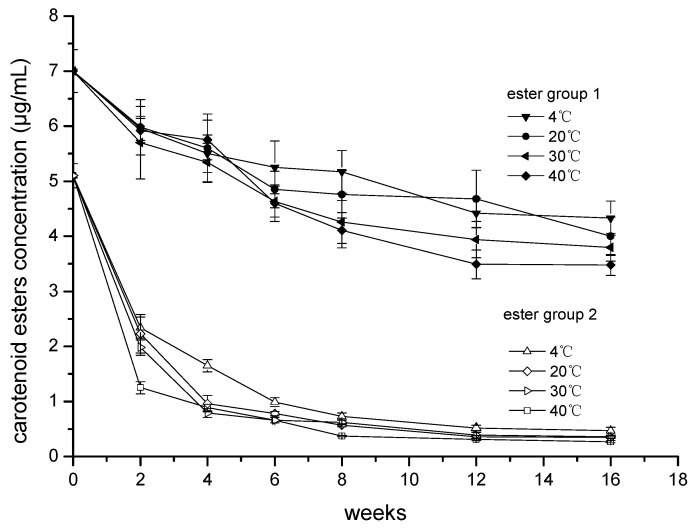
Changes in the content of ester group 1 and ester group 2 (μg/mL) in ‘Cara Cara’ juice during 16 weeks of storage at different temperature.

**Figure 3 foods-08-00417-f003:**
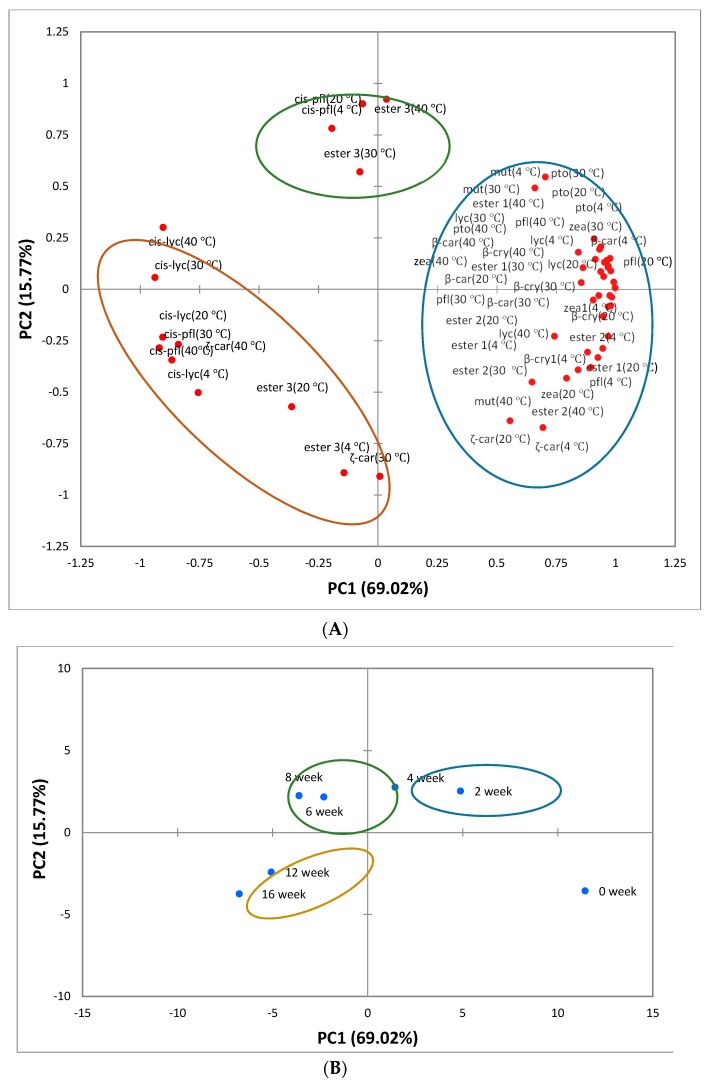
Principle component analysis (PCA) of carotenoid compounds in ‘Cara Cara’ juice during 16 weeks of storage at 4, 20, 30 and 40 °C. (**A**) Loading plot of PCA, (**B**) scores scatter plot of PCA. Note: mut for mutatoxanthin, zea for zeaxanthin, β-cry for β-cryptoxanthin, ζ-car for ζ-carotene, β-car for β-carotene, cis-lyc for 13- or 15-cis-lycopene, lyc for lycopene, pto for phytoene, pfl for phytofluene, cis-pfl for cis-phytofluene, ester 1 for ester group 1, ester 2 for ester group 2, ester 3 for ester group 3. PC1and PC2 represent the first principal component and the second principal component, respectively.

**Figure 4 foods-08-00417-f004:**
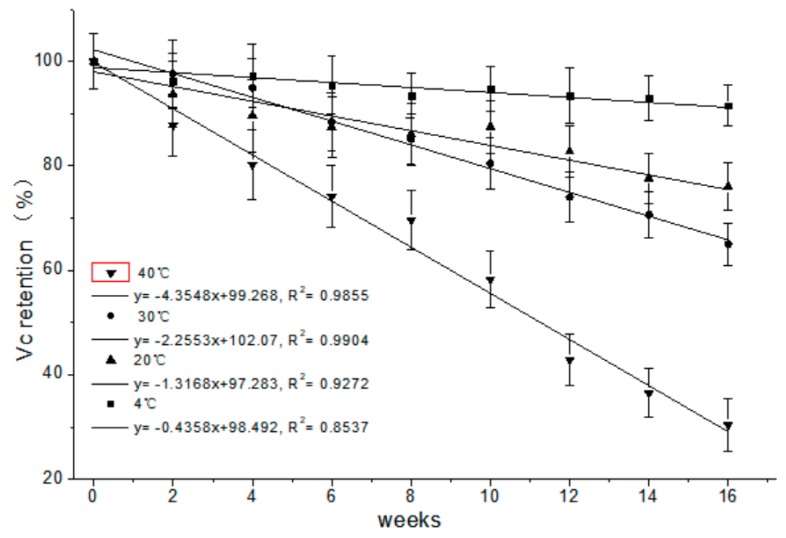
Percent retention of vitamin C in ‘Cara Cara’ juice during 16 weeks of storage at different temperatures.

**Table 1 foods-08-00417-t001:** Changes in the content of flavonoids (μg/mL) in ‘Cara Cara’ juice during 16 weeks of storage at different temperatures.

Weeks	Apigenin-6,8-di-C-Glucoside	Narirutin-4′-O-Glucoside	Narirutin	Hesperidin	Didymin	Total Flavonoid Index
4 °C						
0	30.37 ± 2.47 a	30.43 ± 1.69 a	140.62 ± 6.47 a	620.22 ± 22.86 a	76.15 ± 3.24 a	897.79 ± 36.73 a
4	26.46 ± 0.48 a	27.41 ± 1.93 a	137.64 ± 12.29 a	618.98 ± 29.43 a	76.45 ± 2.67 a	886.94 ± 46.80 a
8	27.48 ± 1.36 a	26.77 ± 2.38 a	136.84 ± 13.91 a	601.63 ± 16.38 a	76.16 ± 5.33 a	868.88 ± 39.39 a
12	25.49 ± 1.56 a	25.93 ± 2.58 a	134.51 ± 6.17 a	583.72 ± 5.90 a	74.34 ± 3.85 a	843.99 ± 20.06 a
16	25.04 ± 0.47 a	25.52 ± 1.42 a	129.25 ± 4.35 a	567.65 ± 13.95 a	72.91 ± 1.34 a	820.37 ± 21.53 a
20 °C						
0	30.37 ± 2.47 a	30.43 ± 1.69 a	140.62 ± 6.47 a	620.22 ± 22.86 a	76.15 ± 3.24 a	897.79 ± 36.73 a
4	25.63 ± 2.35 ab	26.86 ± 3.25 ab	136.82 ± 4.57 a	614.16 ± 8.56 ab	75.28 ± 4.47 a	878.75 ± 23.20 a
8	25.35 ± 1.30 ab	24.66 ± 0.92 ab	133.32 ± 1.82 a	592.5 ± 17.68 ab	72.73 ± 2.45 a	848.56 ± 24.17 a
12	23.37 ± 0.53 b	24.65 ± 0.62 ab	125.89 ± 10.60 a	574.42 ± 10.99 ab	74.45 ± 1.34 a	822.78 ± 24.08 a
16	21.25 ± 1.24 b	21.77 ± 1.45 b	126.78 ± 5.32 a	556.75 ± 12.03 b	71.42 ± 1.38 a	797.95 ± 21.42 a
30 °C						
0	30.37 ± 2.47 a	30.43 ± 1.69 a	140.62 ± 6.47 a	620.22 ± 22.86 a	76.15 ± 3.24 a	897.79 ± 36.73 a
4	27.33 ± 3.03 a	26.35 ± 1.65 ab	133.14 ± 6.59 a	603.28 ± 17.21 ab	74.15 ± 1.27 a	864.25 ± 29.75 a
8	26.25 ± 1.73 a	27.06 ± 1.25 ab	128.52 ± 12.45 a	569.28 ± 8.79 ab	73.10 ± 5.83 a	824.21 ± 30.05 a
12	23.71 ± 0.66 a	22.91 ± 0.92 b	121.33 ± 5.08 a	567.69 ± 10.58 ab	73.92 ± 0.75 a	809.56 ± 17.99 a
16	22.35 ± 2.62 a	23.26 ± 2.98 b	122.57 ± 9.82 a	550.59 ± 16.56 b	70.70 ± 3.23 a	789.47 ± 35.21 a
40 °C						
0	30.37 ± 2.47 a	30.43 ± 1.69 a	140.62 ± 6.47 a	620.22 ± 22.86 a	76.15 ± 3.24 a	897.79 ± 36.73 a
4	24.63 ± 1.45 ab	26.19 ± 1.55 ab	131.68 ± 10.37 a	594.84 ± 14.22 ab	74.07 ± 4.63 a	851.41 ± 32.22 a
8	25.48 ± 0.93 ab	26.58 ± 1.12 ab	132.44 ± 4.75 a	579.33 ± 15.21 ab	72.25 ± 2.24 a	836.08 ± 24.25 a
12	23.36 ± 1.23 b	24.88 ± 1.69 ab	130.66 ± 8.46 a	570.12 ± 11.13 ab	73.33 ± 1.68 a	822.35 ± 24.19 a
16	23.59 ± 2.43 ab	24.09 ± 1.35 b	127.59 ± 10.68 a	552.17 ± 12.54 b	71.29 ± 3.36 a	798.73 ± 30.36 a

Values are expressed as mean ± SD, *n* = 3. Values followed by different letters in the columns are significantly different (Duncan’s multiple range tests, *p* < 0.05). Total flavonoid index represents the sum of individual quantified flavonoid concentrations.

**Table 2 foods-08-00417-t002:** Changes in the content of soluble sugar concentrations (mg/mL) in ‘Cara Cara’ juice during 16 weeks of storage at different temperatures.

Weeks	Fructose	Glucose	Sucrose	Total
4 °C				
0	6.96 ± 0.17 a	15.15 ± 0.90 a	23.22 ± 1.85 a	45.33 ± 2.92 a
2	6.95 ± 0.31 a	15.09 ± 0.30 a	23.16 ± 1.25 a	45.21 ± 1.86 a
4	6.90 ± 0.42 a	14.92 ± 0.28 a	23.52 ± 1.36 a	45.40 ± 2.06 a
6	7.02 ± 0.36 a	15.36 ± 0.44 a	23.04 ± 1.33 a	45.36 ± 2.13 a
8	6.94 ± 0.55 a	15.08 ± 0.54 a	22.89 ± 0.83 a	44.93 ± 1.92 a
10	7.11 ± 0.43 a	15.58 ± 0.65 a	22.68 ± 1.42 a	45.22 ± 2.50 a
12	6.75 ± 0.56 a	14.98 ± 0.36 a	22.74 ± 1.88 a	44.68 ± 2.80 a
14	6.48 ± 0.38 a	15.97 ± 0.45 a	22.88 ± 1.76 a	45.81 ± 2.59 a
16	6.54 ± 0.25 a	15.72 ± 0.23 a	22.02 ± 1.42 a	44.70 ± 1.90 a
20 °C				
0	6.96 ± 0.17 a	15.15 ± 0.90 a	23.22 ± 1.85 a	45.33 ± 2.92 a
2	7.03 ± 0.12 a	15.13 ± 0.14 a	23.94 ± 1.67 a	46.03 ± 1.93 a
4	6.92 ± 0.42 a	15.29 ± 0.64 a	23.38 ± 1.79 a	45.63 ± 2.85 a
6	6.68 ± 0.88 a	15.23 ± 0.92 a	23.06 ± 1.15 a	45.25 ± 2.95 a
8	6.94 ± 0.37 a	15.11 ± 0.78 a	23.55 ± 1.84 a	45.62 ± 2.99 a
10	6.56 ± 0.21 a	15.64 ± 0.44 a	23.44 ± 1.38 a	46.04 ± 2.03 a
12	6.62 ± 0.54 a	15.79 ± 0.59 a	23.16 ± 1.47 a	45.91 ± 2.60 a
14	7.08 ± 0.52 a	16.07 ± 0.84 a	22.43 ± 1.15 a	45.46 ± 2.51 a
16	6.84 ± 0.35 a	16.13 ± 0.72 a	21.50 ± 1.55 a	44.59 ± 2.62 a
30 °C				
0	6.96 ± 0.17 a	15.15 ± 0.90 a	23.22 ± 1.85 ab	45.33 ± 2.92 a
2	6.75 ± 0.29 a	15.16 ± 0.85 a	23.60 ± 1.13 a	45.72 ± 2.27 a
4	6.79 ± 0.50 a	15.71 ± 0.78 a	22.12 ± 2.05 ab	44.79 ± 3.33 a
6	7.14 ± 0.61 a	15.85 ± 0.69 a	21.81 ± 1.28 ab	44.62 ± 2.58 a
8	7.18 ± 0.65 a	16.00 ± 1.36 a	21.03 ± 1.11 ab	43.99 ± 3.12 a
10	7.15 ± 1.85 a	16.20 ± 0.46 a	20.37 ± 0.64 ab	43.53 ± 2.95 a
12	6.88 ± 0.47 a	16.68 ± 0.96 a	19.54 ± 1.57 ab	43.18 ± 3.0 a
14	7.17 ± 0.28 a	16.85 ± 1.27 a	19.00 ± 1.42 ab	42.81 ± 2.97 a
16	7.18 ± 0.17 a	16.05 ± 0.47 a	18.09 ± 1.54 b	41.10 ± 2.18 a
40 °C				
0	6.96 ± 0.17 d	15.15 ± 0.90 c	23.22 ± 1.85 a	45.33 ± 2.92 a
2	8.10 ± 0.24 cd	17.32 ± 1.37 c	20.87 ± 1.59 a	45.15 ± 3.2 a
4	8.69 ± 0.22 bc	18.10 ± 1.04 cb	16.97 ± 1.21 b	42.03 ± 2.47 ab
6	9.05 ± 0.38 abc	21.89 ± 1.53 ab	9.77 ± 0.24 c	38.62 ± 2.15 abc
8	9.04 ± 0.81 abc	22.43 ± 1.38 a	5.82 ± 0.30 d	35.21 ± 2.49 bc
10	10.01 ± 0.66 ab	22.70 ± 0.79 a	4.06 ± 0.11 de	33.72 ± 1.56 bc
12	9.34 ± 0.75 abc	23.43 ± 1.50 a	2.77 ± 0.14 de	33.16 ± 2.39 c
14	9.45 ± 0.38 abc	23.42 ± 1.03 a	1.59 ± 0.06 e	31.88 ± 1.47 c
16	10.78 ± 0.38 a	24.01 ± 1.04 a	nd	30.97 ± 1.42 c

Values are expressed as mean ± standard deviation, *n* = 3. Values followed by different letters in the columns are significantly different (Duncan’s multiple range tests, *p* < 0.05).

**Table 3 foods-08-00417-t003:** Changes of the hydrophilic and lipophilic antioxidant abilities in ‘Cara Cara’ juice during 16 weeks of storage at different temperatures.

Weeks	Hydrophilic ABTS ^1^	Hydrophilic DPPH ^2^	Lipophilic ABTS	Lipophilic DPPH
4 °C				
0	6.42 ± 0.44 a	2.14 ± 0.06 a	0.88 ± 0.03 a	0.76 ± 0.02 a
4	6.23 ± 0.12 a	2.09 ± 0.06 a	0.81 ± 0.02 ab	0.75 ± 0.03 a
8	5.89 ± 0.31 a	2.1 ± 0.06 a	0.87 ± 0.02 a	0.74 ± 0.05 a
12	5.71 ± 0.21 a	2.04 ± 0.06 a	0.80 ± 0.03 ab	0.74 ± 0.04 a
16	5.54 ± 0.24 a	1.94 ± 0.05 a	0.77 ± 0.05 ab	0.72 ± 0.02 a
20 °C				
0	6.42 ± 0.44 a	2.14 ± 0.06 a	0.88 ± 0.03 a	0.76 ± 0.02 a
4	5.84 ± 0.31 a	1.92 ± 0.05 b	0.82 ± 0.04 a	0.73 ± 0.06 a
8	5.89 ± 0.32 a	1.86 ± 0.05 b	0.87 ± 0.06 a	0.69 ± 0.02 a
12	5.81 ± 0.24 a	1.82 ± 0.04 b	0.76 ± 0.03 a	0.67 ± 0.03 a
16	5.48 ± 0.21 a	1.64 ± 0.03 c	0.75 ± 0.04 a	0.67 ± 0.04 a
30 °C				
0	6.42 ± 0.44 a	2.14 ± 0.06 a	0.88 ± 0.03 a	0.76 ± 0.02 a
4	6.19 ± 0.27 a	1.84 ± 0.05 b	0.83 ± 0.06 a	0.74 ± 0.03 ab
8	6.06 ± 0.32 a	1.82 ± 0.03 b	0.82 ± 0.04 a	0.66 ± 0.05 ab
12	5.76 ± 0.28 a	1.64 ± 0.04 c	0.76 ± 0.05 a	0.68 ± 0.03 ab
16	5.53 ± 0.34 a	1.56 ± 0.05 c	0.73 ± 0.03 a	0.62 ± 0.02 b
40 °C				
0	6.42 ± 0.44 a	2.14 ± 0.06 a	0.88 ± 0.03 a	0.76 ± 0.02 a
4	6.35 ± 0.31 cd	1.96 ± 0.06 ab	0.86 ± 0.03 ab	0.72 ± 0.01 ab
8	6.26 ± 0.32 bc	1.80 ± 0.05 bc	0.79 ± 0.02 ab	0.73 ± 0.03 ab
12	6.18 ± 0.24 abc	1.72 ± 0.03 c	0.75 ± 0.05 b	0.64 ± 0.05 ab
16	5.91 ± 0.26 abc	1.61 ± 0.06 c	0.74 ± 0.03 b	0.63 ± 0.04 b

^1^ DPPH: expressed as ascorbic acid equivalent (μmol AAE/mL); ^2^ ABTS: expressed as Trolox equivalent (μmol TE/mL). Values are expressed as mean ± SD, *n* = 3. Values followed by different letters in the columns are significantly different (Duncan’s multiple range tests, *p* < 0.05).

**Table 4 foods-08-00417-t004:** Correlation matrix between flavonoid compositions and antioxidant activity in hydrophilic extracts from stored ‘Cara Cara’ juice.

	Hydrophilic ABTS^+^	Hydrophilic DPPH
apigenin-6,8-di-C-glucoside	0.752 **	0.857 **
narirutin-4’-O-glucoside	0.808 **	0.870 **
narirutin	0.702 **	0.901 **
hesperidin	0.699 **	0.874 **
didymin	0.608 **	0.882 **
total flavonoid index	0.742 **	0.910**

** indicated the correlated factors with each other reached significant level, *p* < 0.01.

**Table 5 foods-08-00417-t005:** Correlation matrix between carotenoid compositions and antioxidant activity in lipophilic extracts from stored ‘Cara Cara’ juice.

	Lipophilic ABTS^+^	Lipophilic DPPH
mutatoxanthin	0.420	0.391
zeaxanthin	0.724 **	0.634 **
β-cryptoxanthin	0.704 **	0.604 **
ζ-carotene	−0.178	−0.195
β-carotene	0.640 **	0.558 *
13- or 15-cis-lycopene	−0.655 **	−0.666 **
lycopene	0.652 **	0.600 **
phytoene	0.698 **	0.619 **
phytofluene	0.760 **	0.848 **
cis-phytofluene	−0.629 **	−0.764 **
ester group 1	0.865 **	0.799 **
ester group 2	0.703 **	0.645 **
ester group 3	−0.225	−0.293
total carotenoid index	0.732 **	0.664 **

* indicated the correlated factors with each other reached significant level, *p* < 0.05. ** indicated the correlated factors with each other reached significant level, *p* < 0.01.

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
