# Peer review of "Stability of Flavonoid, Carotenoid, Soluble Sugar and Vitamin C in ‘Cara Cara’ Juice during Storage"

_foods, 2019, doi:10.3390/foods8090417_

Round 1

Reviewer 1 Report

The manuscript by Lu et al provides a well-designed, conducted and presented study demonstrating the aging of Cara Cara juice during storage at different temperatures. The authors used UPLC and HPLC for determination of juice composition followed by ABTS and DPPH assays for antioxidant capacity determination. In this manuscript, appropriatte statistical analysis were performed in order to present the results and discuss them. The authors demonstrated that the total flavonoid and carotenoid content was not significantly affected within the experiment. Futhermore, the results of antioxidant capacity were positively correlated with the content of the flavonoids and carotenoids.

The manuscript is extremely well presented and was a pleasure to read.

I have some suggestions for further improvement of the manuscript before publication:

As "rpm" (revolutions per minute) is not an useful unit depending on the  radius of machine, I prefer to use "g" as a unit for centrifugation steps, which refers to the acceleration applied to your samples. Units should be written in square brackets. Unify the writting of space in case of p<0.05 (e.g. line 129 versus  line 150). Citation [9] is one scientific report; therefore, it could not be refer as a "several reports". (line 48) Unify the units in 2.5 chapter (mmol - line 95, umol - line 96. I miss the data about calibration curves of AAE and TE - e.g. linear range, R2. Add two spaces in the line 119 and one space in the line 122 (between the value and unit). I miss the threshold values and degree of freedom in case of Pearson correlation coefficient. The variables used for PCA analysis are not obvious. Could it be added to Methodology? Line 147 and 176 should not start by bold letter. Line 211 fix "Ester" for "ester". I miss the statistical analysis (t-test) in Figure 2 and 4. (Table 1) - changes in flavonoid content and the sum of flavonoid. Is it typical to find only 5 flavonoids in the juice? Is the "total flavonoid" correctly used? Is not it just the sum of quantified flavonoids? Why is SD so high in case of ester group 1 and so low in ester group 2 (Figure 2)? Fix spaces in Figure 3: 15.77 % for 15.77% (Table 2, 40 ⁰C) change the order in statistical analysis (from "a" belonging to time "0"). Carefully check the literature according to the Guideline. Line 293: Non-sense to use decimals, I prefer to round the values to integers as the deviation is in range of integers. Table A.1 I miss the basic parameters of this method as e.g. limits of detection, linear range, R2, recoveries and repeatability

Author Response

Point 1: As "rpm" (revolutions per minute) is not an useful unit depending on the radius of machine, I prefer to use "g" as a unit for centrifugation steps, which refers to the acceleration applied to your samples.

Response 1: Thank you for the suggestion, and We have revised the unit accordingly. According to the centrifugation machine parameters, the 8000 rpm was changed as 19360 g (Line 82), and 4000 rpm was revised as 9680 g (line 93) .

Point 2: Units should be written in square brackets.

Response 2: According to the guideline and recently published papers in this journal, all the units were presented in brackets as “( )”, and we have carefully checked the figures and tables.  

Point 3:  Unify the writting of space in case of p<0.05 (e.g. line 129 versus line 150).

Response 3: Thank you for this comment. We have unified the expression of p < 0.05 or p < 0.01 with space throughout the manuscript.

Point 4: Citation [9] is one scientific report; therefore, it could not be refer as a "several reports" (line 48) .

Response 4: We appreciated for your suggestion. “several reports” has been revised as “A previous study”.

Point 5: Unify the units in 2.5 chapter (mmol - line 95, umol - line 96. I miss the data about calibration curves of AAE and TE - e.g. linear range, R2.

Response 5: The units have been unified as μmol. The required parameters for calibration curves of AAE and TE have been added (line: 99-102).

Point 6: Add two spaces in the line 119 and one space in the line 122 (between the value and unit).

Response 6: Done accordingly.

Point 7: I miss the threshold values and degree of freedom in case of Pearson correlation coefficient. The variables used for PCA analysis are not obvious. Could it be added to Methodology?

Response 7: The t test has been added in Line 134. “Carotenoid compounds in Cara Cara juice at all store temperatures were sorted by Principal component analysis (XLSTAT 2016, Addinsoft, New York, NY)” has been added in Line 135-136.

Point 8: Line 147 and 176 should not start by bold letter.

Response 8: Done accordingly.

Point 9: Line 211 fix "Ester" for "ester".

Response 9: Done accordingly.

Point 10:  I miss the statistical analysis (t-test) in Figure 2 and 4.

Response 10: We performed the statistical analysis in Figure 2 and 4. Error bars were presented in those figures, while letters indicating statistical differences were not marked in the figures due to the complicated data presented.

Point 11: (Table 1) - changes in flavonoid content and the sum of flavonoid. Is it typical to find only 5 flavonoids in the juice? Is the "total flavonoid" correctly used? Is not it just the?

Response 11: Based on our previous study, there are about 20 phenolics existed in “Cara Cara”, and many of them are phenoilcs derivatives. Therein, apigenin-6,8-di-C-glucoside, narirutin-4’-O-glucoside, narirutin, hesperidin and didymi are five typical flavonoids in the juice. (Reference: Characterization of phenolics and antioxidant abilities of red navel orange “Cara Cara” harvested from five regions of China. )

The “total flavonoid” was not expressed correctly, and “sum of quantified flavonoids” as suggested is more suitable. Based on a previous study (Reference: Microwave-assisted extraction of phenolics with maximal antioxidant activities in tomatoes), total phenolic index was chosen to represent the sum of quantified flavonoid concentrations. Therefore, “total flavonoid content” was revised as “total flavonoid index” and “Total flavonoid index represents the sum of individual quantified flavonoid concentrations.” was added (Line 154-155). “Total caotenoid index represents the sum of individual quantified carotenoid concentrations” was added in supplementary material.

Point 12: Why is SD so high in case of ester group 1 and so low in ester group 2 (Figure 2)?

Response 12: The SD values were calculated according to the values of three repeats.

Point 13:  Fix spaces in Figure 3: 15.77 % for 15.77%

Response 12: Done accordingly.

Point 14: (Table 2, 40 ⁰C) change the order in statistical analysis (from "a" belonging to time "0").

Response 14: The alphabetical order in Duncan's analysis marks the highest value with letter a.

Point 15: Carefully check the literature according to the Guideline.

Response 14: Done accordingly.

Point 16: Line 293: Non-sense to use decimals, I prefer to round the values to integers as the deviation is in range of integers.

Response 16: The values in this study were relatively low, and rounding the values to integers might not show the deviation.

Point 17: Table A.1 I miss the basic parameters of this method as e.g. limits of detection, linear range, R2, recoveries and repeatability 

Response 17: Carotenoid quantification was performed according to parameters presented in our previous study (Reference: Effect of thermal treatment on carotenoids, flavonoids and ascorbic acid in juice of orange cv. Cara Cara).

Reviewer 2 Report

The findings is worth publishing in this journal. However, there are some gray areas that the author(s) need to clarify:

The use of Cara Cara is confusing. This is a cv. of navel orange and it has to denoted as such by either having cv. Cara Cara or 'Cara Cara'. Otherwise, it confuses the reader. This has to be corrected throughout the text. Line 44: Correct the spelling of carotenoids Line 73: which are these other analytical grade chemicals? specify Line 127: replace shown with presented

Author Response

Point 1: The use of Cara Cara is confusing. This is a cv. of navel orange and it has to denoted as such by either having cv. Cara Cara or 'Cara Cara'. Otherwise, it confuses the reader. This has to be corrected throughout the text.

Response 1: We appreciated for your suggestion, and Cara Cara was revised as 'Cara Cara' throughout the text.

Point 2: Line 44: Correct the spelling of carotenoids

Response 2: Thank you for this suggestion and we have carefully checked the spelling.

Point 3: Line 73: which are these other analytical grade chemicals?

Response 3: Analytical grade chemicals have been added in the revised manuscript. Other analytical grade chemicals such as ethanol, hexane and sodium hydroxide were bought from Sinopharm chemical reagent Co., Ltd (Shanghai, China). Line 71-73.

Point 4: specify Line 127: replace shown with presented 

Response 4: Done accordingly.

Reviewer 3 Report

Line 2-3: Cara Cara, Please mark the scientific name. Line 56-57: Please elaborate on how to save after sample collection, how much sample weight is used for juice extraction and how to make juice step, etc. Line 58-59: How to set the sterilization temperature of juice?. Figure 3: Too much data, can't see clearly Table 2: Data expression cannot be zero, please correct it as not detected.

Author Response

Point 1: Line 2-3: Cara Cara, Please mark the scientific name.

Response 1: As it was suggested by reviewer 2, “This is a cv. of navel orange and it has to be denoted as such by either having cv. Cara Cara or 'Cara Cara'”, and 'Cara Cara' has been used to replace Cara Cara throughout the text.

Point 2:  Line 56-57: Please elaborate on how to save after sample collection, how much sample weight is used for juice extraction and how to make juice step, etc. Line 58-59: How to set the sterilization temperature of juice?.

Response 2: The fresh 'Cara Cara' fruit (50 kg) was immediately peeled and squeezed with a fruit extruder. Half of the crude juice was stored at -80°C and half of them was directly subjected to a rapid thermal sterilization. Specifically, 'Cara Cara' juice was boiled in a stainless steel container with a electronic thermometer (F1, invisible, Guangdong, China) monitoring the internal temperature of 'Cara Cara' juice (98°C, 16 s). The sterilized juice was immediately filled into glass bottles (50 mL). Above contents have been added in Line 59-63.

Point 3: Figure 3: Too much data, can't see clearly

Response 3: Figure 3 has been modified with the data clearly presented.

Point 4: Table 2: Data expression cannot be zero, please correct it as not detected.

Response 4: Done accordingly.
